# Dengue Seroprevalence and Factors Associated with Dengue Seropositivity in Petaling District, Malaysia

**DOI:** 10.3390/ijerph19127170

**Published:** 2022-06-11

**Authors:** Rui Jie Ng, Zhuo Lin Chong, Mohd Hatta Abdul Mutalip, Chiu-Wan Ng

**Affiliations:** 1Department of Social and Preventive Medicine, Faculty of Medicine, University of Malaya, Kuala Lumpur 50603, Malaysia; chiuwan.ng@ummc.edu.my; 2Institute for Public Health, National Institutes of Health, Ministry of Health Malaysia, Shah Alam 40170, Malaysia; m.hatta@moh.gov.my

**Keywords:** community-based study, Malaysia, seroepidemiology, tropical disease, vector-borne disease

## Abstract

Dengue virus (DENV) infection is a major public health concern, posing huge economic and disease burdens globally. In Malaysia, the incidence of DENV infections has increased significantly over the years. Nevertheless, the passive surveillance mechanism applied may not capture the actual magnitude of DENV infection. There was also a paucity of community-based studies exploring DENV seroprevalence. This study aimed to determine the DENV seroprevalence and the associated factors among the urban population in Petaling district, Malaysia. A population-based cross-sectional study was conducted from 18 August to 26 October 2018 with 533 participants recruited. Blood samples were collected and analysed for DENV seropositivity using a composite reference standard comprised of three dengue serological tests. Associated factors were identified by fitting Generalised Linear Mixed Models with binomial error structure and logit link function. DENV seroprevalence obtained was 79.0% (95% CI: 75.2–82.4%). The age-specific DENV seroprevalence showed an increasing trend with advancing age, from 22.7% (95% CI: 9.6–45.0%) for those aged below five years old to 94.9% (95% CI: 81.3–98.7%) for those aged ≥60 years old. Only age group and house level were found to be significant factors associated with DENV seropositivity. The odds of being DENV seropositive generally increased with age, from 13.43 (95% CI: 2.77–65.22) for the 5–9 years old age group to 384.77 (95% CI: 39.27–3769.97) for the ≥60 years old age group, as compared to those aged below 5 years old. For house level, those who lived on the first and second floor (OR: 8.98, 95% CI: 3.16–25.12) and the third floor and above (OR: 4.82, 95% CI: 1.89–12.32) had greater odds of being DENV seropositive compared to those living on the ground floor. This study demonstrated the persistently high DENV seroprevalence among the urban population in Petaling district, which could be useful to evaluate dengue control measures taken and estimate more accurate disease incidence. The associated factors with DENV seropositivity identified could also contribute to undertaking more targeted preventive and control measures.

## 1. Introduction

Dengue virus (DENV) infection is a major public health concern, with an estimated 390 million infections occurring annually [1]. For the past five decades, the incidence of dengue has increased by 30-fold globally, with more than 100 countries currently endemic with dengue [2]. Dengue fever is a mosquito-borne infectious disease caused by the DENV with four serotypes (DENV-1 to DENV-4) from the genus Flavivirus and is transmitted by the *Aedes* mosquitoes. DENV infections may lead to a spectrum of pathological conditions, ranging from mild undifferentiated fever to more severe forms of clinical presentations characterised by haemorrhage and shock [3]. DENV infections pose substantial economic and disease burdens to the affected countries. In Southeast Asia alone, DENV infections result in an estimated annual economic burden of USD 950 million (range: USD 610–1384 million) and an annual average of 214,000 disability-adjusted life years (DALYs) (range: 120,000–299,000 DALYs) [4]. The global incidence of dengue is expected to continue rising due to global warming, rapid urbanisation, and increased international trade and travel [5].

In Malaysia, an integrated strategy for dengue prevention and a control programme has been implemented by the Ministry of Health (MOH) under the National Dengue Strategic Plan since 2011 [6]. Nevertheless, the incidence of DENV infections still increased markedly from 31.6 cases in a population of 100,000 in the year 2000 to 361.2 cases in a population of 100,000 in the year 2014 [6]. Malaysia is now hyperendemic with dengue, where all four DENV serotypes co-circulate and the dominant serotypes alternate over time and geographical locations [7].

Currently, the DENV infections in Malaysia are mainly monitored using a passive surveillance system [7]. The existing legislation makes it mandatory for any suspected or confirmed cases of dengue fever to be reported to the MOH within 24 h [8]. However, the passive notification system in Malaysia is very likely to have underestimated the true incidence of DENV infections due to underreporting [7]. While the existing passive surveillance system may be suitable for monitoring long-term trends or detecting dengue outbreaks, it may not accurately capture the actual magnitude of dengue infections in Malaysia [4,9]. Furthermore, a significant proportion of people infected with DENV are clinically asymptomatic, and hence they are often not captured in the current surveillance system [10].

In its guidelines for the prevention and control of dengue disease, the World Health Organisation recommends that countries supplement their passive surveillance systems with sentinel and active surveillance programmes to accurately determine the disease burden [9]. The incidence of DENV infections can be estimated using a series of seroprevalence data across multiple years based on mathematical modelling comprising prevalence, incidence, and mortality [10]. Nevertheless, inadequate dengue seroprevalence studies are being conducted worldwide, including in Malaysia, especially community-based studies [11,12]. Moreover, there appears to be an epidemiological knowledge gap in Malaysia regarding age-specific dengue seroprevalence data, particularly involving children and teenagers [7,13]. Besides that, understanding the factors associated with DENV seropositivity could assist healthcare policymakers and implementers in planning and taking more targeted preventive and control public health measures in dengue management. Hence, this study aims to determine the seroprevalence of DENV infections and the associated factors with DENV seropositivity among the urban population aged nine months old and above in Petaling district, Selangor, Malaysia.

## 2. Materials and Methods

### 2.1. Study Design

This population-based cross-sectional study was conducted from 18 August to 26 October 2018. This study adhered to the revised 2013 Declaration of Helsinki [14]. Ethics approval was obtained from the Medical Research and Ethics Committee, MOH Malaysia (NMRR-17-853-34393) and the University Malaya Medical Centre Medical Research Ethics Committee (MRECID.NO: 2017426-5171). Written informed consent was obtained from all participants and, where applicable, minor assent.

### 2.2. Study Setting and Study Population

This study was carried out in Petaling district, which has the highest population density and is one of the most urbanised districts in Malaysia (Figure 1) [15]. Petaling district was chosen as the study site because it recorded the highest number of dengue hotspots in the country from 2010 to 2015 [16]. A dengue hotspot is defined as an area with a dengue outbreak for more than 30 days. On the other hand, a dengue outbreak is an area where two or more dengue cases have occurred within a 200 m radius of the index case and 14 days notification period [17,18]. Cluster random sampling was applied to select two communities covered by the District Health Office for dengue vector control activities, each from those with the highest and the lowest cumulative reported dengue cases from 2013 to 2017. Section 7, Shah Alam, was selected to represent communities from the highest category, and Section 10, Petaling Jaya (PJ), was selected from the lowest category (marked with pins in Figure 1).

A total of 500 participants were needed based on the single proportion sample size formula to achieve a <10% margin of error for the expected DENV seroprevalence in these communities with a 5% type I error [11]. The participants were recruited from 250 households sampled randomly from both communities proportionate to their population size, with an expectation of 4 people on average in each household and a 50% attrition rate due to refusal or ineligibility [19]. All residents aged ≥ nine months old who had been living in the selected household for the past six months were included in the study. Those with acute febrile illness, history of other flaviviral infection or vaccination, or contraindications to blood taking were excluded.

### 2.3. DENV Serological Analysis

DENV seroprevalence was determined using three different dengue serological tests as a composite reference standard to detect anti-dengue antibodies, which were (1) in-house hemagglutination inhibition (HI) assay, (2) in-house focus reduction neutralisation test (FRNT) adapted from the plaque reduction neutralisation test (PRNT), and (3) commercially available Panbio Dengue IgG indirect enzyme-linked immunosorbent assay (ELISA) (Abbott, Chicago, IL, USA). HI has been regarded as the gold standard for dengue serological diagnosis due to its high accuracy and ability to determine the exact titre of virus-neutralising antibodies [20,21]. The sensitivity of IgG ELISA is comparable to HI but has greater specificity than HI and is more feasible to be conducted [20,22]. On the other hand, FRNT is the gold standard for the differential serodiagnosis of DENV [23]. As these serological assays exhibit some degree of cross-reactivity with other flaviviral infections such as Japanese encephalitis, Zika, and yellow fever viruses when used individually [24], a composite reference standard was used in this study to overcome the shortcoming and to increase the specificity.

Participants were diagnosed to be DENV seropositive when the blood sample was tested negative on only one out of the three dengue serological tests, which were defined as: (1) for HI (titre of <1:10 as negative and ≥1:10 as positive), and (2) for IgG ELISA (<9 Panbio units as positive, 9–11 Panbio units as equivocal, and >11 Panbio units as negative). For FRNT_95_, the titre is defined as the reciprocal of the highest serum dilution in which viral plaque is reduced by 95% compared to the serum-free control. A titre of <1:10 was regarded as negative against a specific DENV serotype and a titre of ≥1:10 was positive. The final FRNT_95_ positive was defined as a positive test to any of the DENV serotypes.

### 2.4. Data Collection

Information about socio-demographic characteristics and other study variables of interest from all consented participants was obtained by trained researchers through an interviewer-administered questionnaire which was validated (Appendix A). A venous blood sample was also taken from each participant to test for DENV seropositivity. The specimens were collected in plain tubes, chilled, and transferred daily to a virological laboratory at the University of Malaya, Kuala Lumpur, Malaysia. In the laboratory, the specimens were centrifuged, aliquoted, and kept at −80 °C until the tests were performed in batches.

### 2.5. Data Analysis

Descriptive analysis was performed to describe the participants’ characteristics using frequency and percentage. DENV seroprevalence was computed as the number of DENV seropositive cases divided by the number of examined sera. Cross tabulation was carried out to obtain DENV seroprevalence for the independent variables. Afterwards, a univariable analysis was performed by fitting Generalised Linear Mixed Models (GLMM) with binomial error structure and logit link function to assess the crude relation between DENV seropositivity and the independent variables using odds ratio (OR) estimates with 95% confidence interval. All socio-demographic factors, environmental factors, preventive practices, and having good knowledge about *Aedes* biting time were included in the univariable analysis. Household was fitted as the random factor, considering that more than one participant could be recruited from the same household. Variables with *p* < 0.25 from the crude analysis were selected to be included in the multivariable model. Correlations between the selected variables were also determined with strongly correlated variables excluded from multivariable analysis to avoid inflation of standard errors, with a cut-off value of 0.50 used to indicate strong correlation [25]. Significance of the association was determined through the adjusted OR estimates with 95% confidence intervals and *p* < 0.05. All statistical analyses were performed using STATA statistical software version 14.2 (StataCorp, College Station, TX, USA).

## 3. Results

### 3.1. Characteristics and DENV Seroprevalence of the Study Participants

A total of 533 participants were eligible for the study, but only 500 consented and provided blood samples (93.8% response rate). Table 1 shows the general characteristics of the participants. DENV seroprevalence among the urban population in Petaling district obtained from the study was 79.0% (95% CI: 75.2–82.4%).

#### 3.1.1. Socio-Demographic Factors

The median age was 28.3 years old (interquartile range: 27.8 years old). The participants were predominantly male (52.0%), Malays (76.0%), Malaysians (95.6%), single (54.8%), employed (52.6%), received tertiary education (45.0%), and had three to five household members (56.0%). Age-specific DENV seroprevalence generally showed a gradual increasing trend with advancing age, except for the 20–29-year-old and 30–39-year-old age groups. When the two selected communities were analysed individually, the age-specific DENV seroprevalence for the community from the lowest cumulative reported dengue cases category showed an ascending trend, from 0.0% for those aged below 10 years old to 94.3% (95% CI: 79.2–98.6%) for those aged 60 years old and above. A gradual increasing trend was also seen for the age-specific DENV seroprevalence for the community from the highest cumulative reported cases category, from 53.3% (95% CI: 38.7–67.4%) for those aged below 10 years old to 100.0% for those aged 60 years old and above, except for the 10–19-year-old age group, which had a DENV seroprevalence of 90.9% (95% CI: 82.0–95.6%) (Figure 2). The DENV seroprevalence was comparable between males and females (79.6% vs. 78.3%). Comparing Malaysians and non-Malaysians revealed the former had a higher DENV seroprevalence (79.3%). For other socio-demographic factors, the highest DENV seroprevalence was found among participants who were Chinese (80.6%), married (86.5%), received up to secondary level education (85.7%), employed (83.9%), and had one to two household members (87.5%). 

#### 3.1.2. Environmental Factors

For environmental factors, most participants lived in high-rise buildings (81.8%), on the third floor and above (50.7%), and did not have indoor potted plants (89.0%). Participants who lived in high-rise buildings were found to have higher DENV seroprevalence (80.4%). Specifically, those who lived on the first and second floors had the greatest DENV seroprevalence (85.8%). Besides that, those who did not have indoor potted plants had higher DENV seroprevalence (79.6%).

#### 3.1.3. Preventive Practices

Preventive practices carried out by the participants were explored, whereby 53 participants (10.6%) used screened windows, 18 participants (3.6%) used screened doors, five participants (1.0%) used a bed net, 182 participants (36.4%) used a mosquito coil/mat/liquor vaporiser, 411 participants (82.2%) used insecticide aerosol spray, 206 participants (41.2%) used air conditioners at home, 69 participants (13.8%) used mosquito repellent cream/spray, and 179 participants (35.9%) used larvicide. Regarding the elimination of stagnant water indoor and outdoor, 104 participants (20.8%) and 84 participants (16.8%), respectively, did it every day. Those who used screened windows had a higher DENV seroprevalence than those who did not (81.1% vs. 78.8%). The same observation was found for those who used a bed net (100.0% vs. 78.8%), mosquito coil/mat/liquid vaporiser (79.1% vs. 78.9%), air conditioner (80.1% vs. 78.2%), and larvicide (88.8% vs. 78.7%). In contrast, those who did not use insecticide aerosol spray had a greater DENV seroprevalence than those who did (86.5% vs. 77.4%), as well as those who did not use screened doors (79.1% vs. 77.8%) and mosquito repellent cream/spray (79.1% vs. 78.3%). The highest DENV seroprevalence was seen among those who eliminated stagnant water every day, both indoor (82.7%) and outdoor (83.3%).

#### 3.1.4. Knowledge

Most (73.4%) participants did not have good knowledge about *Aedes* mosquito biting time. Participants who had good knowledge about *Aedes* biting time had a higher DENV seroprevalence (85.7%).

### 3.2. Factors Associated with DENV Seropositivity

In the univariable analysis, DENV seropositivity was significantly associated with age group, marital status, education status, occupation, usage of larvicide, and having good knowledge about *Aedes* mosquito biting time (*p* < 0.05) (Table 2). Upon checking for correlation among the independent variables with *p* < 0.25 in the univariable analysis, age group was found to be strongly correlated with marital status (Cramer’s *V* = 0.75), education status (Cramer’s *V* = 0.64), occupation (Cramer’s *V* = 0.66), and usage of larvicide (Cramer’s *V* = 0.56). Similarly, there were strong correlations found between occupation and marital status (Cramer’s *V* = 0.55), house level and ethnicity (Cramer’s *V* = 0.59), and usage of larvicide and education status (Cramer’s *V* = 0.51). Hence, marital status, education status, occupation, usage of larvicide, and ethnicity were not included in the multivariable analysis.

Only age group and house level remained significant factors associated with DENV seropositivity in the multivariable analysis after controlling for confounders (Table 2). Compared to those aged below 5 years old, the odds of being seropositive generally increased with age: 5–9 years old (OR = 13.43, 95% CI: 2.77–65.22), 10–19 years old (OR = 23.81, 95% CI: 5.61–101.12), 40–49 years old (OR = 35.06, 95% CI: 6.62–185.59), 50–59 years old (OR = 94.63, 95% CI: 14.36–623.77), and ≥60 years old (OR = 384.77, 95% CI: 39.27–3769.97). There was, however, a fall in the ascending trend among the 20–29 years old (OR = 12.75, 95% CI: 2.85–57.00) and 30–39 years old (OR = 15.24, 95% CI: 3.13–74.29) age groups as compared to those aged below 5 years old. When age was analysed as a continuous variable in the multivariable analysis, there was a 6.0% higher odds of being DENV seropositive with each increasing year (OR = 1.06; 95% CI: 1.03–1.08). For house level, those who lived at higher floors had greater odds of being DENV seropositive compared to those living on the ground floor.

## 4. Discussion

The study provided an up-to-date community-based DENV seroprevalence among the urban population in Malaysia with the inclusion of participants from nine months old and above, which was lacking [11,12]. These DENV seroprevalence data obtained could play important roles in estimating the true incidence of dengue infections in Malaysia, planning public health interventions, and evaluating the vector control programme [10,26]. Besides that, it could be used in estimating the impact of potential vaccine implementation using mathematical modelling [26,27]. The age-specific DENV seroprevalence data could also influence Malaysia’s decisions and strategies in introducing dengue vaccines [28]. In addition, the associated factors with DENV seropositivity identified in this study could contribute to undertaking more targeted preventive and control measures.

The DENV seroprevalence found among the urban population in Petaling district, which constantly records one of the highest dengue hotspots in Malaysia, was 79.0%. This result was comparable to a more recent study conducted in the year 2019 with a seroprevalence of 74.1%, albeit from a much smaller sample size [29]. Another conveniently sampled study conducted at a private health clinic in Petaling district almost two decades ago saw a DENV seroprevalence of 76.5% [30]. Based on these few available studies, the DENV seroprevalence in Petaling district seemed to consistently hover around the range of 70.0–80.0% over the years. These findings also concurred with the study by Chew et al. (2016), which found that multiple series of DENV seroprevalence data in urban areas observed between 2001 and 2013 did not show any time trend difference and had remained consistently high [11]. However, a national population-based study obtained a much higher DENV seroprevalence of 85.4% [13]. A possible reason could be due to the internal migration of the population moving to Petaling district from various parts of the country with different dengue endemicity because Petaling district is one of the main economic and educational hubs of Malaysia [31]. Population movement could have influenced the DENV seroprevalence results, lowering the seroprevalence rate if many migrants were coming from less dengue-prone areas [32]. Compared with other dengue-endemic Southeast Asian countries, the seroprevalence rate in this study was higher than that found in Singapore [33,34], while countries such as Thailand, Laos, and the Philippines reported even higher rates [35,36]. Compared to Latin American countries, which are also endemic with dengue, the DENV seroprevalence rate was comparable to two other cities, namely São Paulo, Brazil [37] and Maracay, Venezuela [38], but was lower than that of another community study conducted in Columbia [39].

The age-specific DENV seroprevalence in this study demonstrated a gradual increasing trend as age advanced. This was in line with other DENV seroprevalence studies conducted locally and internationally, which found an increment in DENV seropositivity rate with increasing age [1,11,13,26,36,40]. This observation was expected, as the longer a person resides in a dengue-endemic area, the greater the risk one is exposed to dengue infection coupled with the lifelong persistence of anti-DENV antibodies [1,13,34]. In the multivariable analysis, age was unsurprisingly found to be significantly associated with DENV seropositivity.

Nevertheless, an anomaly was noted in the age-specific DENV seroprevalence ascending trend for the community from the highest cumulative reported dengue cases category, where a drop was observed among the 20–29-year-old and 30–39-year-old age groups. This could again be due to the aforementioned population migration factor [32]. Education institutions were located in this community, which saw students within the age group of 20–29 years old migrating from other parts of the country for their studies, potentially influencing the DENV seroprevalence obtained. Likewise, working adults and foreign workers within the age group of 20–39 years old also migrated to this community which housed many offices and factories that offered more employment opportunities. The population migration with people originating from areas with lower dengue rates could have brought down the level of DENV seroprevalence obtained in the community for both age groups. The inclusion criteria for this study, which recruited participants living in the area for the past six months, could be revised in future research with the duration made longer to minimise the potential influence of population migration.

Moreover, migration could explain why the community had one of the highest cumulative reported dengue cases over the years. Three scenarios could possibly explain this: (1) the migrants who were non-immune to the circulating DENV serotypes in the community were of greater vulnerability to DENV infections; (2) the migrants could introduce new DENV serotypes to the community, which may lead to the occurrence of DENV outbreaks among the non-immune local residents; and (3) the migrants were generally found to live in housing conditions which were more susceptible to DENV infections compared to local residents, for example crowded rental accommodation [41,42]. Besides that, the constant influx of non-immune migrant population into an endemic community was found to increase the dengue incidence among children as seen in this study among the community from the highest cumulative reported dengue cases category [43]. This was because the migrants served as the susceptible population in the community who were more likely to be symptomatic, had an increased possibility of hospitalisation and a higher viral load, and were at greater risk of transmitting the disease [42,43,44]. Hence, policymakers and public health practitioners should be aware of the effect of population migration on dengue incidence and anticipate a possible increase in dengue outbreaks. This was in contrast with the community from the lowest cumulative reported dengue cases category, which was a mature residential neighbourhood with low levels of migration and thus had a more gradual increasing trend for the age-specific DENV seroprevalence. It was, however, interesting to note that there was a sharp rise in DENV seroprevalence in the same community between the 10–19-years-old to the 20–29-years-old age groups, from 9.1% to 63.6%. As compared to children and adolescents who mostly stayed at home or in school, those who entered adulthood were very likely to undergo changes in their social behaviour when they gained greater autonomy. Studies have found that human social connections and interactions influence the epidemiological transmission of dengue [43,45].

Besides age, the multivariable analysis found that house level was significantly associated with DENV seropositivity, with those living in high-rise buildings having greater odds of being DENV seropositive. Another Malaysian study obtained a similar finding, with the researchers postulating that high population density in high-rise buildings favoured the transmission of dengue virus [46]. In addition, the majority of those living in high-rise buildings in this study were from Section 7, Shah Alam, which witnessed higher migration activity as explained, and had a greater likelihood of living in housing conditions exposing them to DENV infection. Nevertheless, this association warranted further investigation, as two studies conducted in Singapore found no discernible trend between DENV seropositivity and house levels [33,34].

A few other factors in this study, for instance household size, usage of insecticide aerosol spray, elimination of indoor stagnant water, and having good knowledge of *Aedes* biting time, were found to be significant in the univariable analysis but were not significant in the multivariable analysis. Sociodemographic factors were seemingly more context-specific, as other studies had failed to show clear associations with DENV seropositivity [1,13,40,46,47]. Likewise, numerous other studies did not find significant associations between DENV seropositivity and preventive practices [29,36,46,47,48]. While having good knowledge on *Aedes* biting time may influence the adoption of effective preventive measures, no significant association between the knowledge and DENV seropositivity was found in two other studies [29,48]. As the evidence on the effectiveness of household level interventions seems to be unclear, there may be a need to consider introducing dengue vaccination. However, the only licensed dengue vaccine, Dengvaxia, was found to increase the risk of hospitalisation and severe disease in DENV-naïve individuals, despite being moderately efficacious for seropositive individuals [49,50]. Another newer dengue vaccine candidate, which was developed by Takeda, was found to be efficacious and safe in both dengue-naïve and dengue-exposed adults and children in the recently concluded clinical trial [51,52]. Therefore, DENV seroprevalence study is important for evidence-based recommendations in guiding dengue vaccine implementation [28,53].

There are a few strengths associated with this study. Firstly, the participants included those from the younger age groups, which helped address the epidemiological gap of age-specific seroprevalence [7,13]. Secondly, the selection of the study population, which comprised two communities from both the high and low dengue incidence categories, gave a more accurate depiction of the Petaling district population. Thirdly, the community-based DENV seroprevalence study was more representative of the general population compared to other seroprevalence studies, which were more confined, for example, using sentinel surveillance [11] or samples from hospitals [54] and blood banks [26]. Lastly, DENV seropositivity was determined using a composite reference standard, which helped reduce detecting cross-reactivity with other flaviviruses. The limitation of the study was potential self-reporting bias as information provided by participants was not verified by researchers. The causal relationship between independent and dependent variables could not be established as this was a cross-sectional study. In addition, this study investigated the main effects of individual factors on dengue seropositivity. The interaction between these factors and those not included in this study, such as entomological and meteorological factors, should be explored in future research as DENV transmission is a complex process [55,56].

## 5. Conclusions

This study demonstrated the persistently high DENV seroprevalence among the urban population in Petaling district, Malaysia, despite the implementation of integrated strategies for dengue prevention and control. Similar DENV seroprevalence studies should be repeated in the near future and in other dengue-endemic countries, perhaps on a larger scale, if accurate and affordable rapid diagnostic test kits are available. Besides serving as a performance appraisal of current dengue control measures, having a series of DENV seroprevalence data could help in accurately estimating the disease burden, determining the impact of potential vaccine implementation, and making projections about future disease burden, which would allow the development of better control strategies and public health policies. The possibility of population migration leading to higher dengue incidence and outbreaks as highlighted in this study should be anticipated and investigated further, with the ease of travel leading to greater population mobility. Identifying factors associated with DENV seropositivity could also aid in undertaking more targeted preventive and control measures in line with precision public health.

## Figures and Tables

**Figure 1 ijerph-19-07170-f001:**
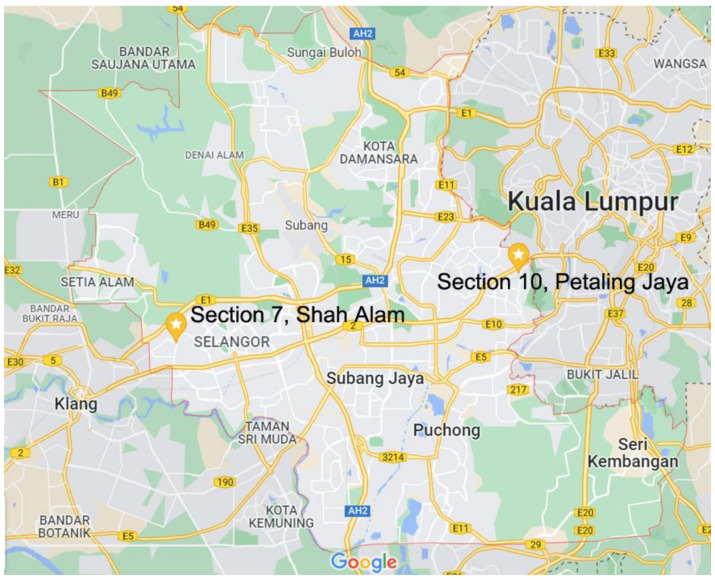
Map of Petaling district, with the two selected communities. (Source: Map data ©2022 Google, with modification).

**Figure 2 ijerph-19-07170-f002:**
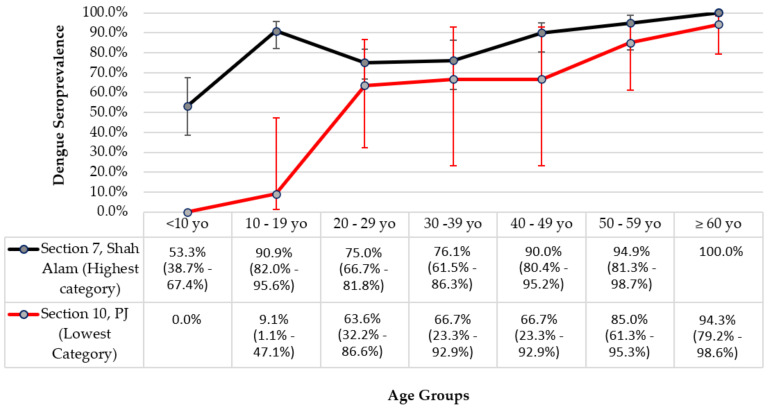
Age-specific dengue seroprevalence with 95% confidence interval by incidence category.

**Table 1 ijerph-19-07170-t001:** Characteristics and DENV seroprevalence status of participants (*n* = 500).

Characteristics	*n* (%)	No. of People DENV Seropositive (% of *n*)	No. of People DENV Seronegative (% of *n*)
**Total**	**500 (100.0)**	**395 (79.0)**	**105 (21.0)**
**Socio-demographic factors**			
Age Group			
<5	22 (4.4)	5 (22.7)	17 (77.3)
5–9	25 (5.0)	19 (76.0)	6 (24.0)
10–19	88 (17.6)	71 (80.7)	17 (19.3)
20–29	139 (27.8)	103 (74.1)	36 (25.9)
30–39	52 (10.4)	39 (75.0)	13 (25.0)
40–49	76 (15.2)	67 (88.2)	9 (11.8)
50–59	59 (11.8)	54 (91.5)	5 (8.5)
≥60	39 (7.8)	37 (94.9)	2 (5.1)
Gender			
Male	260 (52.0)	207 (79.6)	53 (20.4)
Female	240 (48.0)	188 (78.3)	52 (21.7)
Location			
Section 10, Petaling Jaya	91 (18.2)	66 (72.5)	25 (27.5)
Section 7, Shah Alam	409 (81.8)	329 (80.4)	80 (19.6)
Ethnicity			
Malay	380 (76.0)	306 (80.5)	74 (19.5)
Chinese	36 (7.2)	29 (80.6)	7 (19.4)
Indian	54 (10.8)	41 (75.9)	13 (24.1)
Others/Foreigner	30 (6.0)	19 (65.3)	11 (36.7)
Nationality			
Malaysian	478 (95.6)	379 (79.3)	99 (20.7)
Non-Malaysian	22 (4.4)	16 (72.7)	6 (27.3)
Marital Status			
Single/Divorced/Widowed	286 (57.2)	210 (73.4)	76 (26.6)
Married	214 (42.8)	185 (86.5)	29 (13.5)
Education status (*n* = 498)			
Never attended/Primary	92 (18.4)	63 (68.5)	29 (31.5)
Secondary	182 (36.6)	156 (85.7)	26 (14.3)
Tertiary	224 (45.0)	174 (77.7)	50 (22.3)
Occupation (*n* = 483)			
Student	144 (29.8)	109 (75.7)	35 (24.3)
Employed	254 (52.6)	213 (83.9)	41 (16.1)
Unemployed/Homemaker/Retiree	85 (17.6)	65 (76.5)	20 (23.5)
Household size			
1–2	64 (12.8)	56 (87.5)	8 (12.5)
3–5	280 (56.0)	215 (76.8)	65 (23.2)
≥6	156 (31.2)	124 (79.5)	32 (20.5)
**Environmental factors**			
Type of house			
Landed	91 (18.2)	66 (72.5)	25 (27.5)
High-rise	409 (81.8)	329 (80.4)	80 (19.6)
House level (*n* = 499)			
Landed	91 (18.2)	66 (72.5)	25 (27.5)
1st and 2nd floor	155 (31.1)	133 (85.8)	22 (14.2)
3rd floor and above	253 (50.7)	195 (77.1)	58 (22.9)
Indoor potted plants			
Yes	55 (11.0)	41 (74.6)	91 (20.4)
No	445 (89.0)	354 (79.6)	14 (25.4)
**Preventive practices**			
Usage of screened windows			
Yes	53 (10.6)	43 (81.1)	10 (18.9)
No	447 (89.4)	352 (78.8)	95 (21.2)
Usage of screened doors			
Yes	18 (3.6)	14 (77.8)	4 (22.2)
No	482 (96.4)	381 (79.1)	101 (20.9)
Usage of bed net			
Yes	5 (1.0)	5 (100.0)	0 (0.0)
No	495 (99.0)	390 (78.8)	105 (21.2)
Usage of mosquito coil/mat/liquid vaporiser			
Yes	182 (36.4)	144 (79.1)	38 (20.9)
No	318 (63.6)	251 (78.9)	67 (21.1)
Usage of insecticide aerosol spray			
Yes	411 (82.2)	318 (77.4)	93 (22.6)
No	89 (17.8)	77 (86.5)	12 (13.5)
Usage of air conditioner at home			
Yes	206 (41.2)	165 (80.1)	41 (19.9)
No	294 (58.8)	230 (78.2)	64 (21.8)
Usage of mosquito repellent cream/spray			
Yes	69 (13.8)	54 (78.3)	15 (21.7)
No	431 (86.2)	341 (79.1)	90 (20.9)
Usage of larvicide (*n* = 499)			
Yes	179 (35.9)	159 (88.8)	20 (11.2)
No	239 (47.9)	188 (78.7)	51 (21.3)
Do not know	81 (16.2)	47 (58.0)	34 (42.0)
Eliminate stagnant water indoor			
Yes, during epidemics	37 (7.4)	29 (78.4)	8 (21.6)
Yes, every day	104 (20.8)	86 (82.7)	18 (17.3)
Yes, at least once a week	234 (46.8)	193 (82.5)	41 (17.5)
No	125 (25.0)	87 (69.6)	38 (30.4)
Eliminate stagnant water outdoor			
Yes, during epidemics	36 (7.2)	29 (80.6)	7 (19.4)
Yes, every day	84 (16.8)	70 (83.3)	14 (16.7)
Yes, at least once a week	165 (33.0)	134 (81.2)	31 (18.8)
No	215 (43.0)	162 (75.4)	53 (24.6)
**Knowledge (*Aedes* biting time)**			
Good knowledge (*Aedes* biting time) ^1^			
Yes	133 (26.6)	114 (85.7)	19 (14.3)
No	367 (73.4)	281 (76.6)	86 (23.4)

^1^ Good knowledge about *Aedes* biting time was defined as correctly answering early morning and evening as the peak biting periods.

**Table 2 ijerph-19-07170-t002:** Univariable and multivariable analyses of factors associated with DENV seropositivity (*n* = 500).

Characteristics	Univariable Analysis			Multivariable Analysis		
	Crude OR	95% CI	*p*-Value	Adjusted OR	95% CI	*p*-Value
Age Group *						
<5	1.00	Reference		1.00	Reference	
5–9	18.71	3.63–96.35	<0.001	13.43	2.77–65.22	0.001
10–19	27.95	6.66–117.25	<0.001	23.81	5.61–101.12	<0.001
20–29	19.50	4.98–76.33	<0.001	12.75	2.85–57.00	0.001
30–39	20.84	4.80–90.41	<0.001	15.24	3.13–74.29	0.001
40–49	53.63	11.51–249.83	<0.001	35.06	6.62–185.59	<0.001
50–59	81.93	15.47–433.92	<0.001	94.63	14.36–623.77	<0.001
≥60	165.98	20.43–1348.46	<0.001	384.77	39.27–3769.97	<0.001
Gender						
Male	1.00	Reference				
Female	0.81	0.48–1.36	0.429			
Location						
Section 10, Petaling Jaya	1.00	Reference				
Section 7, Shah Alam	1.46	0.70–3.07	0.314			
Ethnicity						
Malay	1.00	Reference				
Chinese	1.01	0.33–3.13	0.985			
Indian	0.75	0.30–1.92	0.556			
Others/Foreigner	0.49	0.16–1.54	0.223			
Nationality						
Malaysian	1.00	Reference				
Non-Malaysian	1.16	0.28–4.81	0.834			
Marital Status						
Single/Divorced/Widowed	1.00	Reference				
Married	2.87	1.62–5.07	<0.001			
Educational Status (*n* = 498)						
Never attended/Primary	1.00	Reference				
Secondary	3.22	1.55–6.67	0.002			
Tertiary	1.85	0.95–3.60	0.072			
Occupation (*n* = 483)						
Student	1.00	Reference				
Employed	2.24	1.17–4.28	0.015			
Unemployed/Homemaker/Retiree	1.01	0.46–2.20	0.988			
Household size						
1–2	1.00	Reference		1.00	Reference	
3–5	0.42	0.16–1.12	0.083	0.66	0.24–1.80	0.420
≥6	0.52	0.18–1.53	0.233	0.66	0.22–2.00	0.463
Type of house						
Landed	1.00	Reference				
High-rise	1.46	0.70–3.07	0.314			
House level (*n* = 499)						
Ground	1.00	Reference		1.00	Reference	
1st and 2nd floor	2.36	0.97–5.74	0.059	8.98	3.16–25.12	<0.001
3rd floor and above	1.13	0.52–2.46	0.750	4.82	1.89–12.32	0.001
Indoor potted plants						
Yes	0.77	0.31–1.88	0.560			
No	1.00	Reference				
Usage of screened windows						
Yes	1.00	Reference				
No	0.72	0.27–1.94	0.513			
Usage of screened doors						
Yes	1.00	Reference				
No	1.23	0.25–6.12	0.802			
Usage of bed net						
Yes	NA ^1^	-				
No						
Usage of mosquito coil/mat/liquid vaporiser						
Yes	1.00	Reference				
No	1.05	0.57–1.91	0.881			
Usage of insecticide aerosol spray						
Yes	1.00	Reference		1.00	Reference	
No	2.13	0.96–4.70	0.062	1.96	0.85–4.52	0.116
Use of air conditioner at home						
Yes	1.00	Reference				
No	0.89	0.48–1.63	0.696			
Usage of mosquito repellent cream/spray						
Yes	1.00	Reference				
No	1.10	0.51–2.37	0.813			
Usage of larvicide (*n* = 499)						
Yes	1.00	Reference				
No	0.54	0.28–1.03	0.062			
Do not know	0.15	0.07–0.32	<0.001			
Eliminate stagnant water indoor						
Yes, during epidemics	1.00	Reference		1.00	Reference	
Yes, every day	1.16	0.38–3.56	0.789	1.02	0.33–3.22	0.967
Yes, at least once a week	1.09	0.40–2.97	0.867	1.13	0.41–3.15	0.811
No	0.44	0.15–1.26	0.127	1.21	0.37–3.96	0.756
Eliminate stagnant water outdoor						
Yes, during epidemics	1.00	Reference				
Yes, every day	1.41	0.43–4.64	0.574			
Yes, at least once a week	0.95	0.32–2.78	0.925			
No	0.57	0.20–1.65	0.299			
Good knowledge (*Aedes* biting time)						
Yes	1.00	Reference		1.00	Reference	
No	0.50	0.27–0.94	0.031	0.59	0.30–1.18	0.135

* When age was analysed as a continuous variable in multivariable analysis, the OR obtained was 1.06 (95% CI: 1.03–1.08) with *p*-value < 0.001. NA: not available. ^1^ omitted due to perfect prediction.

## Data Availability

The data presented in this study are available on request from the corresponding author. The data are not publicly available due to agreement of confidentiality.

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
