# Peer review of "Dengue Seroprevalence and Factors Associated with Dengue Seropositivity in Petaling District, Malaysia"

_ijerph, 2022, doi:10.3390/ijerph19127170_

Round 1

Reviewer 1 Report

The aim of the study  was to determine the seroprevalence of DENV infections 75 and the associated factors with DENV seropositivity among the urban population in 76 Petaling district, Selangor, Malaysia.

The conclusions of the conducted research are clear and result from the obtained research results. The material used for the research is sufficient, the research methods have been selected appropriately. The arrangement of the figure and tabels is clear and presents the obtained results very well. Discussing the results against the background of other authors is very detailed. The publications cited by the authors of the article are well selected. For the most part, the authors refer to the latest knowledge published in renowned scientific journals. I could not find any mistakes in the scientific aspect of the manuscript.

However, the authors did not avoid a few mistakes, which I will list below:

- A few punctuation problems are present in the manuscript. I suggest the Authors to double-check the text.

Reviewer 2 Report

Dear authors,

your paper "Dengue Seroprevalence and Factors Associated with Dengue 2 Seropositivity in Petaling District, Malaysia" is very interesting and has many strengths. I have some suggestions for you:

  1. Introduction part - I think you must improve this part. First of all, separate introduction and literature review part. At the end of the introduction, explain the aims of your paper. It would be great to mention research gaps that your paper solved. Also, put references for these constatations - DENV infections may lead to a spectrum of 40 pathological conditions, ranging from mild undifferentiated fever to more severe forms 41 of clinical presentations characterized by hemorrhage and shock. Also, you must add more references regarding your research.
  2. In the methods part, please add the next parts: study area with map; sample; Please, explain how you have conducted face to face interviews
  3. Results part - from the results part move the sample to the methods part; Also, try to present your results in tables and figures, there are a lot of data in the text; Also, please separate descriptive and inferential statistics results. 
  4. Conclusion part is poorly written, please add research and social implications of your research; also, don't repeat results in your conclusion part;

Please, add your research instrument at the end of the paper.

Kind regards

Reviewer 3 Report

Please, see the document attached.

Reviewer 4 Report

The authors conducted a cross-sectional study in Petaling District, Selangor, Malaysia. They aimed to know the seroprevalence of dengue infection and determine its main driver factors. The issue of the reviewed manuscript is relevant and it counts on international interest, therefore strongly recommended to publish after a revision.

Major notes:

The manuscript is well structured and written, albeit with questionable statistical analysis. The study is based on 500 participants who lived in 250 households. Mathematically it means the average of sampled participants was two people per household (at least one and could be more than two). Important considerations when conducting logistic regression are the compliances of assumptions. Perhaps the most important assumption is independence. This is to ensure that observations at the outset of the study are not probabilistically related (see Stoltzfus, 2011; Denis, 2019). In this case, I think the observations appear in groups (households); thus, the independence may be violated. The people who live there can be characterised by similar environmental factors and very similar preventive practices. This approach can overestimate or underestimate some drivers and can cause misinterpretation. The authors, therefore, should refine their statistical analysis.

The authors explained the anomaly in age-specific DENV seroprevalence in the highest community with the migration of the 20-29-year-old and 30-39-year-old people. The seroprevalence rose boldly in the mentioned age categories. The authors should explain what could be in the background of the opposite changing of seroprevalence in the different communities. It might be possible, but they detected an opposed phenomenon in the lowest community.

In the case of some variables, too small number of outcomes was gained. These variables (other ethnicity, widowed/divorced, retiree as occupation, bed net use...) should be analysed together with similar ones (e.g. retirees with homemakers) or remove (e.g. bed net use is a very rare phenomenon among investigated persons). Separate analysis of these outcomes may bias the results.

Only one reference to collinearity examination could be found (Table 2, footnote 2). For example, in the case of ethnicity and nationality the number of both foreigners and non-Malaysians is 22. It is suspected that the two numbers mean the same group of people. Use of air-conditioner mostly depends on economic conditions, which can be influenced by education status or even by ethnicity. These correlations should be analysed. The method of analysis should also be published.

The combination of different factors can also be interesting because they do not affect alone. For this reason, a supplementary data collection is suggested to publish.

Minor notes:

In Line 40, 172, 215, 219, Table 1 and Table 2, the Aedes is italic as it is a genus name.

References:

Stoltzfus, J.C., 2011. Logistic regression: a brief primer. Acad Emerg Med, 18(10), 1099-104.

Denis, D.j., 2019. SPSS data analysis for univariate, bivariate, and multivariate statistics. John Wiley and Sons, Inc.

The manuscript provides valuable epidemiological information on Dengue virus infection. Some methodological correction (see above) will improve its general quality. The findings of the manuscript are expected to generate further research activities.

Round 2

Reviewer 2 Report

Dear authors,

Thank you for your corrections.

Kind regards